# Time-Series of Cloud-Free Sentinel-2 NDVI Data Used in Mapping the Onset of Growth of Central Spitsbergen, Svalbard

**Stein Rune Karlsen** [1,*] **, Laura Stendardi** [2] **, Hans Tømmervik** [3] **, Lennart Nilsen** [4] **, Ingar Arntzen** [1] **and Elisabeth J. Cooper** [4]

1   NORCE Norwegian Research Centre AS, P.O. Box 6434, 9294 Tromsø, Norway; inar@norceresearch.no
2   Department of Agriculture, Food, Environment and Forestry (DAGRI), University of Florence, Piazzale delle Cascine, 18-50144 Firenze, Italy; laura.stendardi@unifi.it
3   Norwegian Institute for Nature Research (NINA), FRAM—High North Research Centre for Climate and the Environment, P.O. Box 6606, Langnes, 9296 Tromsø, Norway; Hans.Tommervik@nina.no
4   Department of Arctic and Marine Biology, UiT—The Arctic University of Norway, 9037 Tromsø, Norway; lennart.nilsen@uit.no (L.N.); elisabeth.cooper@uit.no (E.J.C.)
*   Correspondence: skar@norceresearch.no; Tel.: +47-934-19904

**Abstract:** The Arctic is a region that is expected to experience a high increase in temperature. Changes in the timing of phenological phases, such as the onset of growth (as observed by remote sensing), is a sensitive bio-indicator of climate change. In this paper, the study area was the central part of Spitsbergen, Svalbard, located between 77.28°N and 78.44°N. The goals of this study were: (1) to prepare, analyze and present a cloud-free time-series of daily Sentinel-2 NDVI datasets for the 2016 to 2019 seasons, and (2) to demonstrate the use of the dataset in mapping the onset of growth. Due to a short and intense period with greening-up and frequent cloud cover, all the cloud-free Sentinel-2 data were used. The onset of growth was then mapped by a NDVI threshold method, which showed significant correlation ($r^2 = 0.47$, $n = 38$, $p < 0.0001$) with ground-based phenocam observation of the onset of growth in seven vegetation types. However, large bias was found between the Sentinel-2 NDVI-based mapped onset of growth and the phenocam-based onset of growth in a moss tundra, which indicates that the data in these vegetation types must be interpreted with care. In 2018, the onset of growth was about 10 days earlier compared to 2017.

**Keywords:** Sentinel-2; NDVI; time-series; onset of growth; Svalbard

## 1. Introduction

Global temperature is increasing, and particularly so in Svalbard [1–3]. This has strong impacts on terrestrial ecosystems. Whilst the 1982–2015 greening trend for the Arctic is most pronounced at very high latitudes in continental tundra regions, maritime areas including the high Arctic islands (here defined as oceanic to moderately continental regions), show weaker trends [4–6].

In order to monitor climate-induced change in vegetation, time-series of optical satellite data can be used to map a range of different biophysical parameters, and such data have been used in Svalbard to map phenological stages such as the onset and peak of the growing season, as well as plant productivity [6–8]. A strong link between the plant productivity and sea ice distribution was found [9]. Cloud detection is the most crucial step during the pre-processing of time-series of optical satellite images. Failure to mask out (remove) the clouds from the image will have a significant negative impact on any subsequent analyses, such as differentiating the phenological stages [8,10], change detection and land-use classification [11]. Cloudiness in Svalbard, as well as in the other arctic islands, has caused problems when monitoring the different phenological stages using satellite data [7,9]. Snow and ice, frequent events with fog, low solar elevation angles, short growing seasons, and weak vegetation responses characterize Svalbard, and make

cloud masking on the archipelago a challenging task. Further, Svalbard is undergoing dramatic climatic changes, with periods of heavy rain instead of clear cold weather with steady snow cover in autumn–early winter, and mild periods in midwinter, creating ice- or even snow-free spots that occurred several times in the last few years [12]. How these extreme climatic events and changes in snow duration, snow properties, and time of green-up affect the plant growth is largely unknown at the plant community and ecosystems levels, and the provision of more accurate proxies of phenology at large spatial scales is necessary [13].

Extensive datasets of ground-based visual observations of phenology have been collected in Adventdalen, Svalbard since 2007 [14] and elsewhere in arctic and alpine areas over the last 30 years through the International Tundra Experiment (e.g., [15–17]). However, these are very time-consuming to collect and require constant presence in the field, and so an automated system of observations would be more efficient. It is a challenge to make the links between sensors at different distances from the vegetation, from near-scale up to satellite-derived data [18]. Clearly, there is an urgent need for methodologies to be developed that combine field-based observations and data using near-ground sensors such as those obtained from phenocams [10,19], in order to validate satellite data, e.g., Sentinel-2 (S2) [13,20–22] and provide more accurate estimates of plant phenology and productivity at different spatial scales [13].

Compared with previous studies of the onset of growth in Svalbard based on 231.65 m pixel resolution in MODIS data [7] S2 data with 10–20 m pixel resolution offers exciting opportunities to monitor and study plant phenology at much greater detail [23]. Due to the northern location of Svalbard, and the polar orbit of S2, the study area has frequent acquisition, with often two images of S2 data per day, and is therefore a suitable area for vegetation phenology studies.

The main aim of this study is thus to (1) prepare, analyze and present a cloud-free time-series of daily S2 NDVI datasets for the 2016 to 2019 seasons covering central parts of Spitsbergen in Svalbard, and (2) to demonstrate the use of the dataset in mapping the onset of vegetation growth.

## 2. Materials and Methods

### 2.1. Study Area

The study area was the central part of Spitsbergen (Nordenskiöld Land peninsula, and Nathorst Land) in the archipelago of Svalbard, located between approximately 77.28°N–78.44° N and 12.40°W–19.10°W (Figure 1), which is S2 tile 33XWG and most of tile 33XVG.

Nordenskiöld Land is characterized by large valleys with dense vegetation cover, where the mean NDVI value locally reaches above 0.5 (Figure 1), indicating high plant biomass [8,24]. At higher elevations, the vegetation is sparse, as shown in NDVI values below 0.1 (Figure 1). Nathorst Land is mainly covered by glaciers, and vegetation is only found along the coast and in small valleys. Altogether, 1889 km$^2$ (20.7%), of the land area has NDVI values above 0.1; the remaining area is made up of glaciers or is very sparsely vegetated.

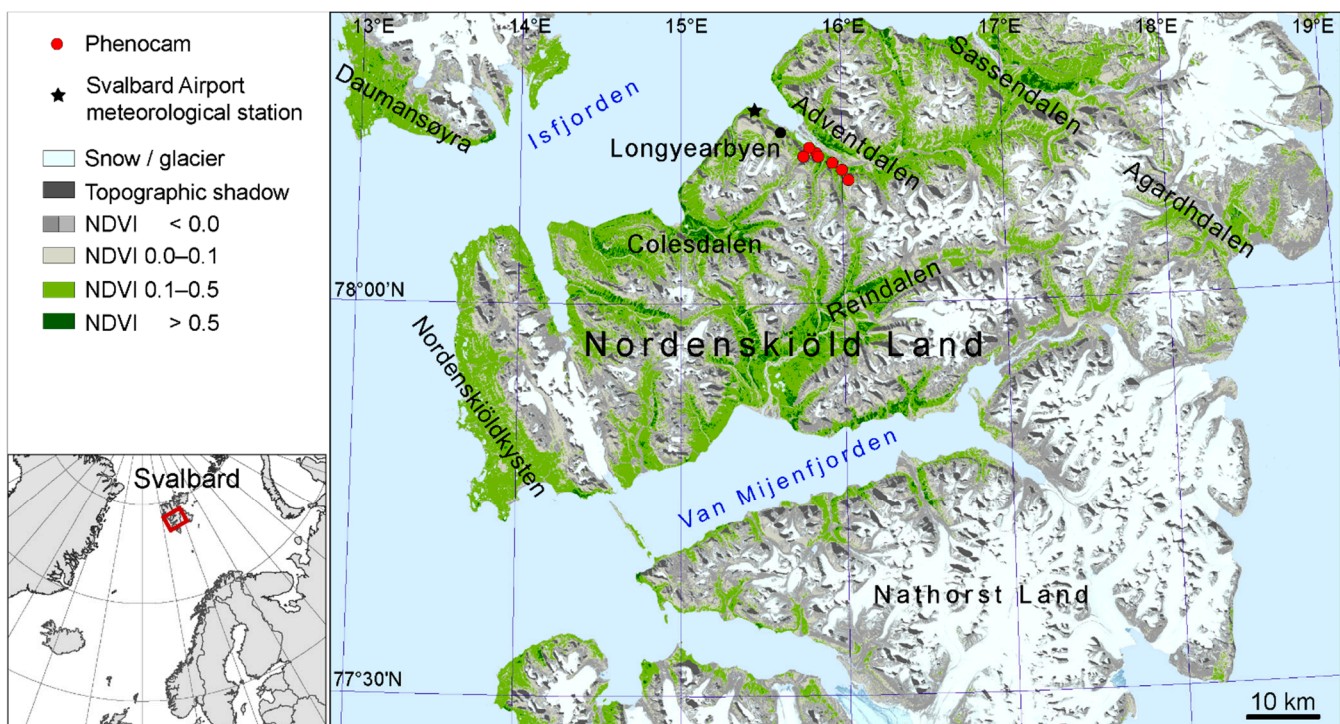

**Figure 1.** The study area of central Spitsbergen, Svalbard, showing mean NDVI values for the 25 July to 1 August period (4-year mean (2016–2019)). Snow/glacier mean values are shown for the same period, extracted from NDSI values (see text). The topographic shadow is approximate when the S2 passes in late July. The map also shows the location of the phenocams used in this study.

The meteorological station at Svalbard Airport, located close to the archipelago's administrative center in Longyearbyen (Figure 1), recorded a mean July air temperature of 7.0 °C and mean annual temperature of −3.8 °C for the 1991−2020 normal period (Table 1). This study is based on use of S2 data for the years 2016 to 2019. For Nordenskiöld Land, the snow melted early in both 2016 and 2018 compared to the 2000–2019 average [6], reflected by high May temperatures of 1.4 °C in 2016 and 1.8 °C in 2018, compared to the −2.2 °C average from the last 30 years (Table 1). On the other hand, the year 2017 had a May temperature of only −3.9 °C. The onset of growth starts in June in the warmest valleys, and in early July at higher elevations [7]. All the years examined in this study (2016–2019) had mean June temperatures above the 1991–2020 average.

**Table 1.** Monthly and annual temperature (°C) for the meteorological station at Svalbard Airport (station number 99840), located close to Longyearbyen (Figure 1). Temperatures for the years with S2 data used in this study compared to the 1991–2020 normal period (Norwegian Meteorological Institute 2021).

| Month/Year | 2016 | 2017 | 2018 | 2019 | 1991–2020 |
|---|---|---|---|---|---|
| May | 1.4 | −3.9 | 1.8 | −2.3 | −2.2 |
| June | 5.0 | 4.6 | 4.0 | 4.8 | 3.6 |
| July | 9.0 | 6.9 | 7.2 | 8.4 | 7.0 |
| Annual | −0.1 | −2.2 | −1.8 | −3.4 | −3.8 |

### 2.2. Phenological In-Situ Data

Phenological observations designed to be up-scaled by S2 data were established in Adventdalen and adjacent Endalen valleys (Figure 1). In the field, large homogeneous areas of the following vegetation types were located: *Dupontia fisheri* marsh, *Dryas octopetala* tundra, mixed *Dryas octopetala-Cassiope tetragona* tundra, moss tundra (*Aulacomnium*

*turgidum-Tomentypnum nitens* type), *Equisetum arvense* ssp. *alpestre* snowbed, and *Luzula confusa* tundra. The mixed *Dryas octopetala-Cassiope tetragona* tundra site was located in Endalen, which has a slightly warmer local climate and represents the warmest/earliest site for growth onset. In addition, we also monitored phenology on a mixed exposed tundra type on a mountain plateau (350 m.a.s.l.). All the selected areas are homogeneous, except the site at the mountain plateau, which is moss rich and where gravel covers about one third, and *Luzula confusa* and *Salix polaris* are the most common vascular plants. Together, these vegetation types make up a large part of the vegetation diversity and thereby the phenological variation in the Adventdalen valley and on a nearby mountain plateau. The smallest site, a very homogeneous *Equisetum* snowbed where the moss *Sanionia uncinata* dominates the ground layer, covers fourteen $10 \times 10$ m$^2$ S2 pixels (1400 m$^2$) and the other six sites from 73 to 361 pixels (7300–36,100 m$^2$). The phenological observations were obtained by manual examination of photos captured by time-lapse cameras (phenocams) (trail camera model Acorn LtL-5310WA with 12-Megapixel and 100° wide angle) placed on tripod 40–60 cm above the surface, covering a plot of 0.95–1.43 m$^2$. The cameras were used in the selected vegetation types at the same site each year, established after snowmelt but before green-up, and removed from the field in September. The phenocams captured images each hour from 10 a.m. to 2 p.m. each day. In the moss tundra and in the mixed *Dryas-Cassiope* tundra we used two cameras, while the other sites had one camera.

In each site, the images capture photos of the dominant and/or the sub-dominant species, except the moss tundra, where both *Dryas octopetala* and *Salix polaris* only occurs scattered on dryer parts (Figure 2). The shrub *Salix polaris* is among the most widespread vascular species in Svalbard, and is very common or even sub-dominant in all the observation sites, except in the *Dupontia fisheri* marsh, and *Salix polaris* was included in some of the images on the other sites as well.

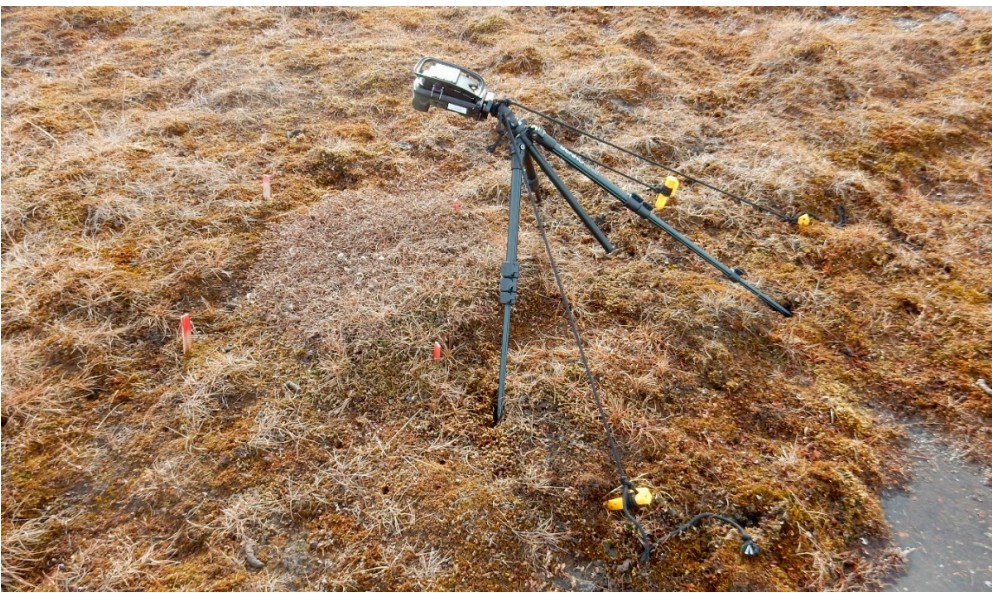

**Figure 2.** A phenocam on a tripod in Adventdalen valley established to capture images of *Dryas octopetala* in a moss tundra. Photo from 3 June 2016, eight days before the defined onset of growth of *Dryas octopetala* that year.

Within the phenocam images, we counted and observed leaves/steams close to a stick or at the same spot each year, since the camera had the same placement each year. The area of interest examined was about $5 \times 5$ cm and had at least 5 leaves of the study species.

From the close-up phenocam images, we followed the phenological growth stages, and we defined the phenophases according to the extended BBCH scale [25]. The abbreviation BBCH derives from the names of the originally participating stakeholders: "Biologische

Bundesanstalt, Bundessortenamt und CHemische Industrie". The BBCH scale is a system for the uniform coding of phenologically similar growth stages of all mono- and dicotyle-donous plant species. Similar phenological stages of each plant species are given the same code. In the two-digit code, the first number shows the principal growth stage (0–9) and the second number the secondary stage (0–8). In this study, we manually extracted the date (day of the year) on which the plant reached the BBCH scale 15, which is on the shrubs we observed: '>5 leaves unfolded, but not yet full size', and the corresponding phenophase on the graminoids we observed: '>5 leaves (>3 cm) clearly visible'. For the horsetail *Equisetum arvense* ssp. *alpestre*, no BBCH code exists, so for this species we defined the onset of growth as: '>5 stem (>3 cm) having branches (>0.5 cm)'. The BBCH scale was established primarily for agricultural plants, but BBCH code 15 gives a good definition of the first opening of leaves of our study species. This phenological stage is called 'onset of growth' in this study and corresponds to 'green-up' in many tundra publications (e.g., [15]). It happens rapidly and is easily observed by eye in the field in Svalbard.

*2.3. Processing Sentinel-2 Data—Cloud Removal and Interpolation*

This study makes use of all available S2 imagery from April/May to mid-September and spans four seasons (2016 to 2019). Svalbard's location close to the North Pole as well as the polar orbit of S2 enables a higher number of captured images. On 1 July 2017, the satellite Sentinel-2A was joined by its twin, satellite Sentinel-2B, and from this point onwards we often have two images per day.

At the same time though, the high latitude implies low solar elevation angles (solar zenith angles higher than 70°) for its measurements, and particularly so late in the growth season. For this reason, processing level 2A (bottom-of-atmosphere (BOA)) is not reliable for this area, as it results in the under-correction of the atmospheric signal [26]. The study is therefore based on processing level 1C (top-of-atmosphere (TOA)) reflectance data. Level 2A data are only used as reference data for scene classification and the estimation of cloud probability.

Overcoming cloud cover is a crucial step during the pre-processing of time-series of optical satellite images. For cloud detection, we examined the cloud probability in Level 2A (Sen2Cor processor), from the 's2cloudless' machine-learning-based algorithm [27], and from previous S2-based algorithms [28]. In addition, we developed our own cloud detection algorithms from multi-spectral values and multi-temporal tests, using experiences from MODIS data time-series processing of Svalbard [7], as, for instance, the algorithm: the value of band 8 (NIR) for a given pixel is compared to its median value for that pixel in the 2016–2019 period, and values less than 30% of the median indicated cloud shadows covering vegetated areas [28]. However, none of the cloud detection methods work well for sparsely vegetated areas (bright surfaces), which is very common in the study area. Additionally, for thin semi-transparent clouds, and for cloud shadows, the algorithms did not show sufficient accuracies. To detect clouds, we performed a visual inspection in the visual and SWIR bands, and visually masked out cloud free areas, only using all the different cloud masks as references. One exception was the cirrus clouds, which could be accurately detected with the S2 band 10. However, cirrus clouds only appear in a few of the images. This time-consuming method for overcoming cloud cover, based on visual inspection, ensured few errors, but not all cloud-free data are included. Some of the images had many small cumulus clouds. Areas scattered with such clouds were too time-consuming to mask out and were therefore not used.

Normalized Difference Vegetation Index (NDVI) values were calculated for the cloud-free pixels and interpolated to daily data. The gap-filling to daily data was achieved by performing a linear interpolation and smoothing with a Savitzky–Golay filter [29]. Areas with NDVI <0.1 (4-year (2016–2019) mean NDVI for the 25 July to 1 August period, Figure 1) were not included in the dataset, as they have no, or at most, very sparse vegetation cover. The interpolation and smoothing method used led to some unrealistically low NDVI values early in the season when it was snow covered, and all NDVI values below −0.1 were

reclassified to −0.1. Finally, due to the short and intense growth season in the study area, we believed that data with more than 10 days since last cloud-free observation would be too inaccurate to be used, and were removed from the dataset. The resulting dataset is daily clear-sky NDVI maps with $10 \times 10$ m$^2$ resolutions for the 2016 to 2019 seasons.

In addition, from the clear-sky S2 dataset, the NDSI (Normalized Difference Snow Index) values were calculated and interpolated to daily data for all land areas in the study area. This S2-NDSI dataset is used in a study where different snow mapping methods are compared [30], and the mean snow cover for 25 July to 1 August (mean 2016–2019) was extracted and used on the maps in this study for improved illustrations.)

### 2.4. Mapping the Onset of Growth

To map the onset of growth each year, we used the time-series of cloud-free daily S2-NDVI data. First, we computed the 4-year (2016–2019) mean NDVI value for every pixel in the study area for the 10 July to 5 August period. This period was chosen as the period where the leaves are fully open and before senescence starts and reduced the "noise" from snow-covered ground. The onset of growth at each pixel each year was defined as the time when the NDVI value each year exceeded 69% of the 10 July to 5 August 4-year mean NDVI value. This NDVI threshold level was reached after several iterations as the level that gives the highest correlation to—and least bias with—the 'onset of growth' observed using phenocam photos. Variations of this method were first used in Fennoscandia [31–35] and were used in Svalbard on MODIS data [7].

## 3. Results

### 3.1. A Clear-Sky Daily NDVI Dataset

The availability of cloud-free data from May to September limits the possibility to map the onset of growth. In 2016, we used S2 data for the 30 April to 12 September period. For this 135-day period, S2 images were captured on 111 days. From these 111 days with S2 data, we obtained on average only 8.2 days with cloud-free data (Figure 3), indicating a cloud cover of greater than 90% of the time in which the satellite captured images that season. In total, 17.5% of the pixels had more than 10 days since the last cloud-free day, and so were removed from the dataset, leading to main gaps in the time-series 9–11 May, 2–3 June, 25–28 June and 21–22 July. In particular, Agardhdalen, some parts of Adventdalen, and south-westernmost parts of the study area had limited S2 data (<8 cloud free days) in the 2016 season, creating restrictions to the reliable mapping of the onset of growth in parts of the study area.

In 2017, we used data for the 9 May to 12 August period; for these 95 days, we had on average 10.5 days with cloud-free pixels. In the west (Nordenskiöldkysten) there was little cloud-free data (less than 8 days), but the main valleys (Reindalen, Adventdalen, and Sassendalen) had about 13–16 days with cloud-free pixels (Figure 3). Some main gaps in the data were 19 June and in the 2–12 July period, and altogether, 16.8% of the pixels had more than 10 days since the last cloud-free day and were removed.

In 2018, S2 data from 30 April to 23 August was processed. However, for this 115-day period, on average, only 7.6 days of cloud-free data were found, and these cloud-free pixels were unevenly spread: while Reindalen, Sassendalen and Agardhdalen had mostly less than 7 days with cloud-free pixels, Daumansøyra had more than 13 days (Figure 3). Due to the cloudy conditions in this season, a total of 21.9% of the pixels had more than 10 days since the last cloud-free day, and this led to large gaps in the dataset between 25 May and 7 June, and from 14 to 19 July.

For the year 2019, we used S2 data in the period from 30 April to 15 August. For these 107 days, we obtained cloud-free data on an average of 18.8 days. In parts of Reindalen, up to 29 days had cloud-free data (Figure 3). Only 4.2% of the pixels had more than 10 days since the last cloud-free day, leading to few gaps in the time-series, and these gaps were mainly in the northwestern parts of the study area between 29 June and 16 July.

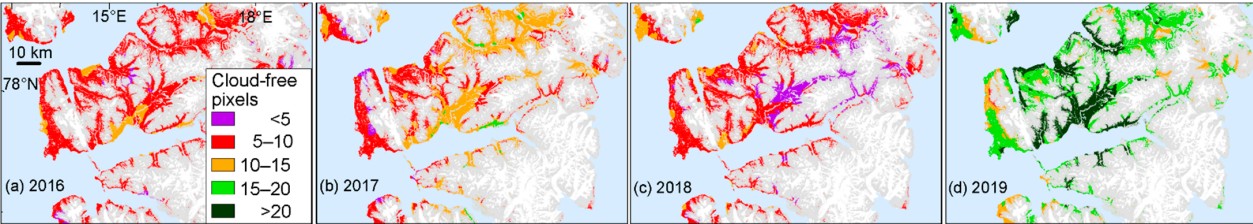

**Figure 3.** Central Svalbard. Number of cloud-free pixels for the season: (**a**) 2016; (**b**) 2017; (**c**) 2018; (**d**) 2019.

### 3.2. Timing of the Onset of Growth

The calculation of the S2-NDVI-based onset of growth for each of the seven polygons, representing the seven vegetation types with phenocams, were compared with the phenocam-based records of onset of growth (Table 2). The correlation between these two methods was highly significant (Figure 4), and on average, the NDVI-based onset of growth occurs one day later than the phenocam-based (day of year 171 vs. 170, 19 vs. 20 June).

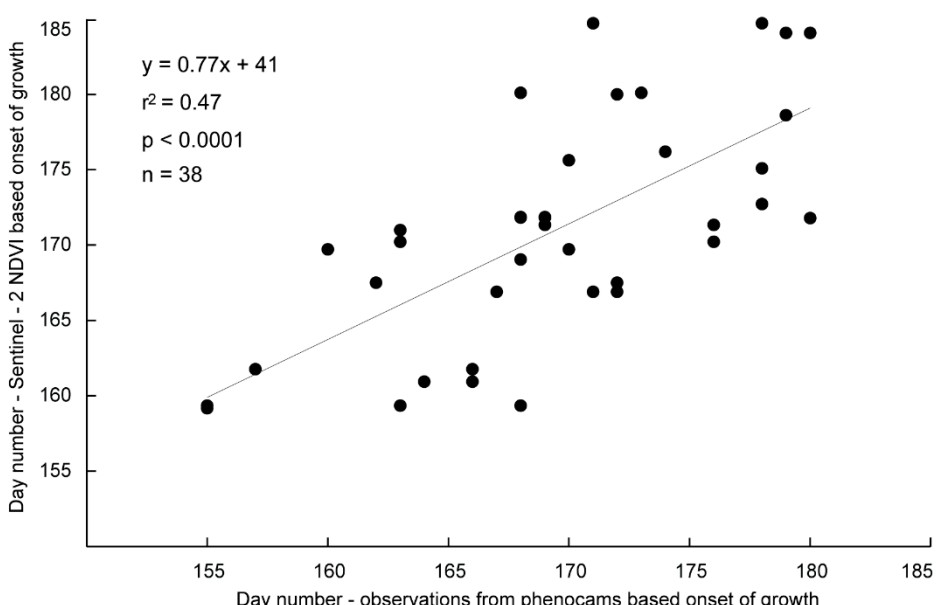

**Figure 4.** Relationship between the phenocam-based observations of onset of growth and the S2-NDVI-based onset of growth.

Some data are missing in Table 2. *Dupontia fisheri* was monitored in field only in 2016, and *Equisetum arvense* ssp. *alpestre* only in 2018 and 2019. The other missing data is due to technical problems with the cameras, or because the cameras fell down during the season.

The earliest onset of growth recorded by phenocams was on *D. octopetala* in 2018 (doy 155/4 June), in the *D. octopetala–C. tetragona* tundra in Endalen, and this polygon also had the earliest S2 NDVI-based onset (doy 159/8 June). In general, the latest onset of growth measured both by phenocam and S2 NDVI was found in 2017. Among the seven vegetation types, there were some differences between species and years in the onset of growth, as observed by phenocams. In general, *Luzula confusa* in the mixed exposed tundra on the mountain plateau, and *E. arvense* in the snowbed site had a late onset of growth compared with the other species/sites, while *D. octopetala* in the mixed *D. octopetala–C. tetragona* tundra in the relatively warm Endalen had an early onset of growth, followed by *D. octopetala* in the *D. octopetala* tundra (Table 2).

**Table 2.** Onset of growth (day of year) as measured from phenocams (left value), and from S2 NDVI (right value) for the seven polygons representing seven vegetation types, see location in Figure 5.

| Vegetation Types/Species | 2016 | 2017 | 2018 | 2019 |
|---|---|---|---|---|
| 1. Mixed exposed tundra | | | | |
| *Luzula confusa* | 178/175 | 179/179 | 174/176 | |
| 2. *Equsetum arvense* snowbed | | | | |
| *Equsetum arvense* ssp. *alpestre* | | | 180/172 | 178/173 |
| 3. *Dupontia fisheri* marsh | | | | |
| *Dupontia fisheri* | 170/176 | | | |
| 4. *Dryas octopetala* tundra | | | | |
| *Dryas octopetala* | 160/170 | | 155/159 | 162/168 |
| *Salix polaris* | 170/170 | | | 172/168 |
| 5. *Luzula confusa* tundra | | | | |
| *Luzula confusa* | 168/172 | 168/180 | 164/161 | 168/172 |
| *Salix polaris* | 169/172 | 173/180 | 166/161 | 169/172 |
| 6. Moss tundra | | | | |
| *Dryas octopetala* | 163/170 | 171/185 | 157/162 | 168/169 |
| *Salix polaris* | 176/170 | 178/185 | 166/162 | |
| 7. Mixed *D. octopetala*–*C. tetragona* tundra | | | | |
| *Dryas octopetala* | 163/171 | 172/180 | 155/159 | 167/167 |
| *Betula nana* | 169/171 | 179/184 | 168/159 | 171/167 |
| *Salix polaris* | 176/171 | 180/184 | 163/159 | 172/167 |

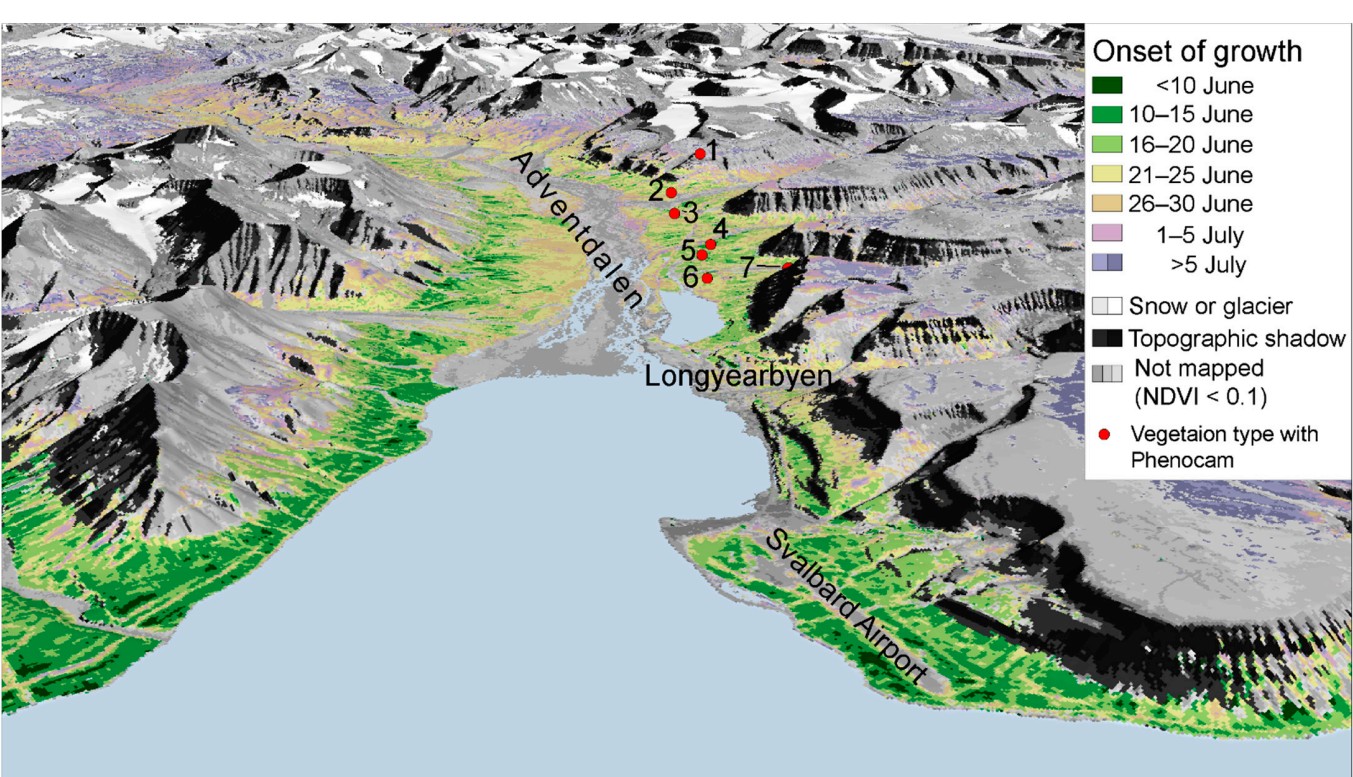

**Figure 5.** Longyearbyen–Adventdalen valley area seen from northwest towards southeast. Showing mean onset of growth for the 2016 to 2018 period, and the placement of the phenocams.

The largest bias and variability between the phenocam and NDVI-based onset of growth were found in the moss tundra, where the scattered occurrences of vascular plants only contribute very little to the NDVI signal. In the moss tundra, the phenocam-observed onset of growth in vascular plants occurs up to 14 days earlier and up to 5 days later compared with the S2-NDVI-based onset.

A large bias of timing was also seen in the *Equisetum* snowbed, where the moss *Sanionia uncinata* dominate the ground layer, which had phenocams in 2018 and 2019, and where the phenocams showed 8- and 5-day later onset of growth, respectively, compared with the S2 NDVI-based onset (Table 2).

### 3.3. Onset of Growth

Figure 5 shows a 3D view of the mean (2016–2019) onset of growth of the Longyearbyen–Adventdalen valley area and illustrates the level of detail used for mapping the onset of growth. An early average onset of growth (<15 June) is found at the coast and in areas with favorable slope (e.g., facing S or SW) and exposure at lower altitudes. The map also reveals a clear altitude gradient, with the onset of growth in July at higher altitudes.

In 2019, the cloud-free S2 data availability was high, and the onset of growth could be mapped in most of the study area (Figure 6d), while in other years (2016–2018), extensive areas could not be mapped for growth onset due to gaps in the S2 NDVI time-series during the spring period (Figure 6a–c). The mapping reveals large differences in the onset of growth between the years. The year 2018 had an early onset, mostly before 15 June at lower altitudes. The year 2016 had an early onset of growth in the western parts, but more average in the eastern parts of the study area. The year 2017 had a very late onset, in early July in most of the study area, except along the coast and in parts of Sassendalen and Agardhdalen, which had an onset of growth in mid-June. In 2019, the onset of growth was close to the 2016–2019 average. In Adventdalen valley, the onset of growth was more than 10 days earlier in 2018 compared to 2017, while at higher altitudes, there were much lower differences between these years.

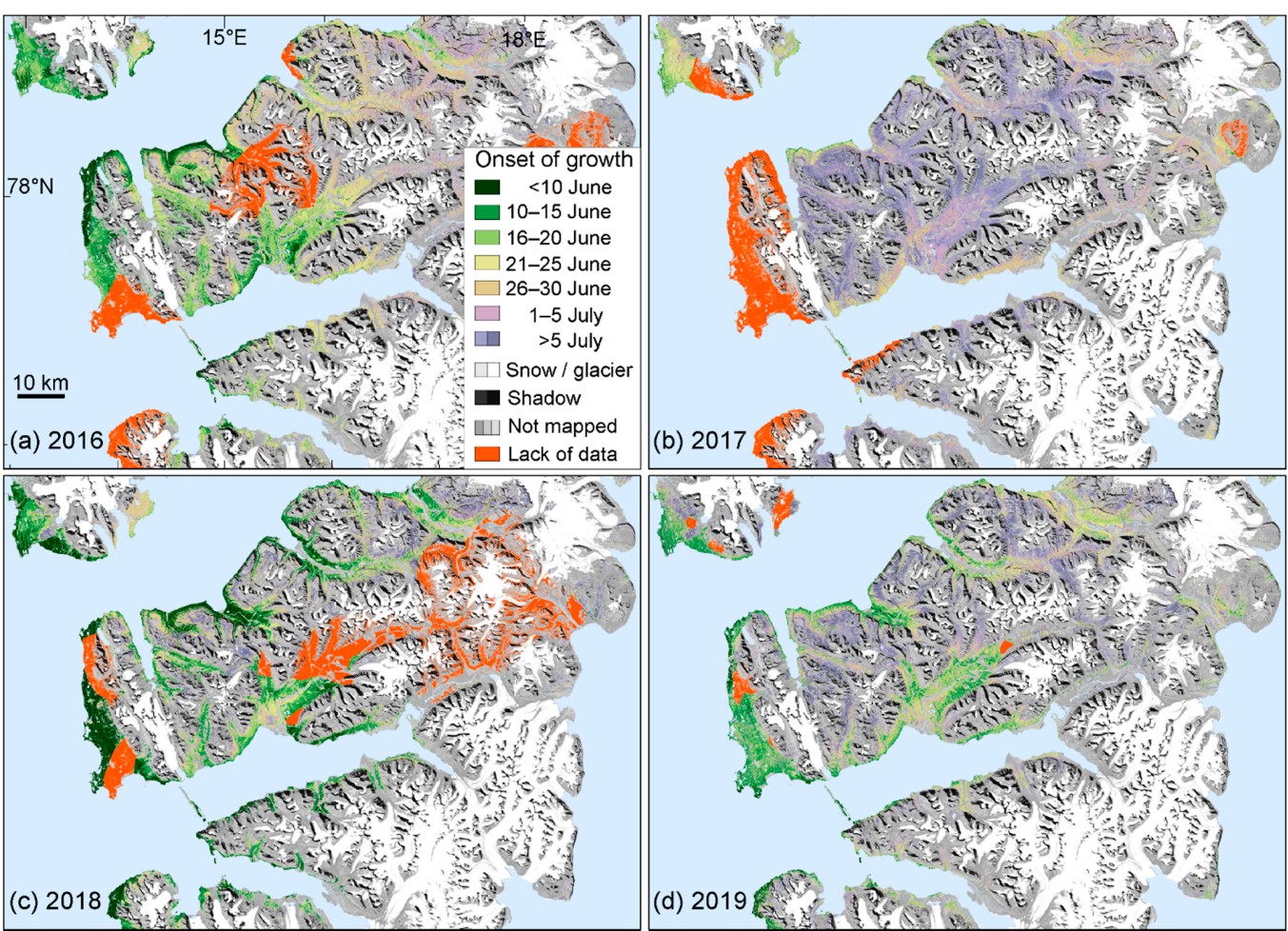

**Figure 6.** Central Svalbard. Onset of growth: (**a**) 2016; (**b**) 2017; (**c**) 2018; (**d**) 2019.

## 4. Discussion

Svalbard has a short growing season, and when green-up occurs, it progresses very rapidly. Hence, it is important to use all cloud-free data to gain an accurate date for the onset of growth. The 2018 season had above-normal cloud cover, and on average only 7.2 cloud-free days were found, which is marginal for processing a NDVI curve. In 2018, the dataset had many gaps (>10 days since last cloud-free pixel), and relying on cloud detection algorithms to clean the dataset would lead too to many errors to create a NDVI curve, at least for most parts of the study area. On the other hand, for the 2019 season, more than 20 cloud-free days were found in some parts of the study area, and for these areas in the 2019 season, we believe cloud detection algorithms could be used, since a suitable gap-filling and smoothing method would 'erase' most of the errors in the cloud masking. Further, including Landsat-8 data [36], which has a close to daily revisit time in the study area, would increase the temporal sampling and reduce the need for visual inspections of cloud cover. In particular, in 2018, the year with less cloud-free data, even after removing large gaps in the NDVI time-series (>10 days since last cloud-free pixel), some errors are found, occurring as unrealistic transitions in the onset of growth maps (Figure 6c).

Previously, Karlsen et al. [8] found that the peak NDVI on Svalbard had much less correlation with plant productivity compared to an integrated NDVI value from the onset to peak of growth, due to the cloudy weather with rarely cloud-free pixels during the time of peak NDVI. This is also the case for this dataset.

Several methods for extracting phenophases from NDVI time-series exist [23]. This study used a NDVI threshold method to detect and map the onset of growth. In high Arctic

areas with snow cover for most of the year and a long period with polar night, methods with the use of the NDVI data only for the snow-free period were shown to be the most accurate [7]. This study showed that the method performs well for all vegetation types studied except the moss tundra.

The NDVI time-series could be improved by including atmospheric correction (from top-of-atmosphere to bottom-of-atmosphere) dealing with the low solar elevation in the study area [26]. Further, NDVI is sensitive to changes in the solar angle, and correcting the dataset with a bidirectional reflectance distribution function (BRDF) would give more accurate NDVI values. However, by including atmospheric correction and BRDF correctionwould only to some degree influence the NDVI trend [37], but the threshold value for estimating the onset of growth then must be adjusted.

The extraction of phenophases from phenocam images on tripods works well in most cases. A few of the cameras fell down during the season, probably kicked down by reindeer. The exact interpretation of the phenocam photos of unfolded leaves on the shrubs *Salix polaris* was difficult due to its small size. However, due to the rapid growth in spring, the period from first leaf development to full-size leaf is often less than two weeks, and by comparing images between the years, to ensure equal interpretation, we believe the inaccuracy in the timing of the phenophase BBCH code 15 [25] used in this study is less than 4 days, and this is in accordance with Anderson et al. [10], who showed a very rapid and similar development for *Salix polaris* in 2016 using both phenocams and Decagon NDVI sensors (8–20 June). Moreover, visual observations in the field in Adventdalen in 2015 at 60 permanent plots gave dates of green-up within a similar timeframe [14], supporting the findings presented here.

We had two phenocams situated in a moss tundra, with only scattered occurrences of vascular plants. These few vascular plants contribute insignificantly to the NDVI values, and we found large bias between the S2 NDVI-based onset of growth and the phenophase BBCH code 15 observed in field on vascular plant. This is an indication that the moss tundra does not directly follow the phenophases of the vascular plants, and here, Anderson et al. [10] showed that the moss tundra had a later greening than vegetation types dominated or characterized by vascular plants such as *Salix polaris*, *Dryas octopetala*, *Cassiope tetragona* and *Luzula* spp. Hence, the separate definition of growth season stages should be defined for the moss tundra, and is an important theme for a future study.

Large differences between years in the date of onset of growth were found using both satellite and phenocam methods. It is not within the remit of the data presented in this paper to explain the annual differences, but it is highly likely that the date of snowmelt and the cumulative temperatures experienced by the plants play an important role in this [14].

## 5. Conclusions

1. In Svalbard, cloud, fog, and haze, in combination with a low sun elevation angle during the short growing season, hinders the acquisition of time-series of ground reflectance data. Cloud detection is hence the most crucial step during the pre-processing of time-series of optical satellite images from such areas.
2. Cloud detection algorithms in S2 data have proven to perform poorly in sparsely vegetated areas (bright surfaces) which are widespread in the study area, and when thin semi-transparent clouds or cloud shadows are present. Thus, additional visual inspection of the visible and SWIR bands was applied to mask-out cloud-free data and ensure as few errors as possible.
3. Normalized Difference Vegetation Index (NDVI) values were calculated for the cloud-free pixels, and interpolated to daily data. A close to complete time-series of daily cloud-free S2 NDVI data could be processed for the 2019 season. For the other years studied (2016–2018) there are several gaps in the time-series. The removal of these pixels resulted in spatial gaps in the onset of growth map.
4. Ground based time-lapse cameras (phenocams) were used within seven vegetation types, and the date of a precisely defined phenophase "onset of growth" with an

accurate botanical definition (BBCH code 15) was extracted from the phenocam images.

5. By applying an NDVI threshold method on the clear-sky time-series of S2 data, the mapping of "onset of growth", shows a significant correlation ($r^2 = 0.47$, n = 38, $p < 0.0001$) with timing of onset of growth as defined from the phenocam images.

6. However, in moss tundra where vascular plants play an insignificant role for the NDVI value, this correlation showed large bias; hence, a separate definition of growth season stages should be defined for the moss tundra.

7. The S2 NDVI-based mapping of onset of growth reveals large differences between the years. In 2018, the onset of growth was more than 10 days earlier compared with 2017, except at higher altitudes. The data presented in this paper are not sufficient to explain these differences, but a future study will examine the relationship between the timing of snowmelt and early growing season temperature.

**Author Contributions:** Conceptualization, S.R.K.; methodology, S.R.K. and L.S.; software, S.R.K., L.S. and I.A.; validation, S.R.K., L.S. and L.N.; formal analysis, S.R.K.; investigation, S.R.K.; resources, S.R.K., H.T. and L.N.; writing—original draft preparation, S.R.K., H.T., L.N., I.A. and E.J.C.; writing—review and editing, S.R.K., H.T. and E.J.C.; visualization, S.R.K.; project administration, S.R.K.; funding acquisition, S.R.K., H.T. and L.N. All authors have read and agreed to the published version of the manuscript.

**Funding:** This work (S.R.K. and L.N.) was supported by the Research Council of Norway under the project Svalbard Integrated Arctic Earth Observing System—Infrastructure development of the Norwegian node (SIOS-InfraNor Project No. 269927). This SIOS project (InfraNord instrument #52) is funded by the Norwegian Space Agency (NoSA). H.T. was supported by European Commission Research and Innovation Action project CHARTER no. 869471.

**Institutional Review Board Statement:** Not applicable.

**Informed Consent Statement:** Not applicable.

**Acknowledgments:** We are grateful to senior advisor John Richard Hansen, at the Norwegian Polar Institute, who in 2010–2016 initiated and supported satellite-based phenological monitoring on Svalbard as a part of the Environmental Monitoring of Svalbard and Jan Mayen (MOSJ). That previous work gave valuable experience for this study. Copernicus Sentinel-2 data were retrieved from ESA SciHub.

**Conflicts of Interest:** The authors declare no conflict of interest.

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
