# Peer review of "Time-Series of Cloud-Free Sentinel-2 NDVI Data Used in Mapping the Onset of Growth of Central Spitsbergen, Svalbard"

_remotesensing, doi:10.3390/rs13153031_

Round 1

Reviewer 1 Report

This paper presents a cloud-free Sentinel2 dataset, and an analysis using those data to calculate the start of seasonal vascular vegetation growth. The methods for creating the data set were rigorous and time consuming, involving a lot of hand cloud-masking. Gap-filling and smoothing methods were used as well as comparison with Landsat-8 data to maximize data availability, and pixels with large data gaps > 10 days of the growing season) were eliminated. This resulted in a 2016-2019 time series of cloud-free data  from Central Svalbard that will be useful for a variety of applications.

The authors used an NDVI threshold method to calculate the onset of growth during the 4 years of the time series. These results correlated well with ground measurements from photocams (interpreted by eye).

The paper is clear and very well written. Minor editorial suggestions follow.

Line 75 – change “suited” to “suitable” or maybe “prime”

Line 98 – change “on -3.8 °C” to “of -3.8 °C”

Line 119 – change “placed in Endalen” to “located in Endalen”

Line 155 – give the range of the BBCH scale

Line 157 – change “on the shrubs we observed on:” to “on the shrubs we observed:”

Line 202 – change “was reclassified” to “were reclassified”

Line 285 – change “was found” to “were found”

Line 301 - change “illustrates level of detail” to “illustrates the level of detail”

Line 395 – change “the time-series have several gaps in the time-series.” to ” there are several gaps in the time-series. Removal of these pixels resulted in spatial gaps in the onset of growth maps.”

Line 407 – change “data presented in this paper is” to “data presented in this paper are”

Reviewer 2 Report

The article is an original study and contains a combination of field data monitoring and satellite observations.

Among the comments and recommendations:

  1. The name of the moss requires correction: "Sanonia uncinata" (lines 128 and 289) to «Sanionia uncinata».
  2. We recommend combining the data from Table 2 (line 279) and Figure 4. Colors and type of markers can to present data by years of work and types of plant communities. This will allow the data to be presented more clearly.
  3. In the text it is desirable to decipher the abbreviation "BBCH scale".

The work can be published.

Reviewer 3 Report

Comments to the Author

This paper describes two works: 1) development of a cloud-free time-series of daily Sentinel-2 NDVI dataset for 2016 to 2019 growing seasons and 2) use this dataset for mapping the onset of growth of central Spitsbergen, Svalbard. This work evaluated the correlation between NDVI-based and phenocam-based onset of growth. Changes in the timing of phenological phases, such as onset of growth, are a sensitive bio-16 indicator of climate change. And the Arctic region is sensitive to climate change. Thus this work is useful to better understand the vegetation dynamic over the study area. However, I do have some main and minor concerns.

Main concerns:

  1. These works seems to combine two works and both of them are not clearly described. The details of cloud detection and data interpolation are not very clear. I expect to see some examples to show what kinds of pixels were detected as cloud pixel and how the interpolation was used. Moreover, the details of the onset of growth calculation both for NDVI-based and ground-based are not clear too. You may also add a figure to show how this process works.
  2. The NDVI dataset contains the sun-sensor geometry effect (BRDF effect) which will add uncertainties into the calculation of the onset of growth. You may do a BRDF correction on the S2 reflectance. Moreover, I think using the TOA is not good for vegetation dynamic study.
  3. Four years are too short for a time series analysis. And you should explain why some data were missing in Tab. 2.

Minor concerns:

Page 1.

L40: Does the sentence after "however" show the same meaning with line 46? I suggest bringing the latter to the front to explain the need for cloud processing on the island.

Page 2.

L71-76: I think this paragraph should be placed before the fourth paragraph of the introduction to make the logic smoother.

Page 4.

L121-123: "which tends to start growth onset later than the other sites (see Table 2 for an overview of species recorded at each site)." This sentence is an explanation of the monitoring results. I think it should be put in the results section.

Page 5.

L155: The abbreviated vocabulary "BBCH" should be commented.

L182: "we developed our own cloud detection algorithms". Please describe your algorithm in more detail. In line 184 of the paper, it says "However, none of the methods work well for sparsely vegetated areas", so what are the advantages of using the method you developed?

L202: This sentence reads wrong. And why did you choose 10 days as the threshold for data filtering? What is the basis for choosing 10 days?

Page 6.

L208: There is an extra ")" here.

L219: You should describe more about this threshold level.

L225: Has the data set been made public?

Page 9.

L308: In Figure 5, the color of the data earlier than June 10 is somewhat overlapped with the terrain shadow.

Round 2

Reviewer 3 Report

  1. The NDVI dataset contains the sun-sensor geometry effect (BRDF effect) which will add uncertainties into the calculation of the onset of growth. You may do a BRDF correction on the S2 reflectance. Moreover, I think using the TOA is not good for vegetation dynamic study.
  2. Four years are too short for a time series analysis. And you should explain why some data were missing in Tab. 2.

The above two main concerns were not well explained in the revised MNS. Since these two problems may affect the reliability of our conclusions. I suggest that you explain these not only to me but also in the text to readers.

You replied that "We agree that BOA is better than TOA, however, due to the low sun elevation most of the season at such northern location (78°N), BOA should not be used. We also processed BOA, and some of these images shows strange effects and cannot be used."

This makes me even more worry about the reliability of the results. Moreover, the very low sun elevation makes it more serious of the BRDF effects. I still suggest that you should first do a BRDF correction using the most common linear kernel-driven BRDF model.

Thus, I suggest another round of major revisions.
